# Exploring Architectural Ingredients of Adversarially Robust Deep Neural Networks

**Hanxun Huang[1]  Yisen Wang[2,3]  Sarah Erfani[1]  Quanquan Gu[4]**
**James Bailey[1]  Xingjun Ma[5]†**
[1]School of Computing and Information Systems, The University of Melbourne, Victoria, Australia
[2]Key Lab. of Machine Perception, School of Artificial Intelligence, Peking University, Beijing, China
[3]Institute for Artificial Intelligence, Peking University, Beijing, China
[4]University of California, Los Angeles, USA
[5]School of Computer Science, Fudan University, Shanghai, China

## Abstract

Deep neural networks (DNNs) are known to be vulnerable to adversarial attacks. A range of defense methods have been proposed to train adversarially robust DNNs, among which adversarial training has demonstrated promising results. However, despite preliminary understandings developed for adversarial training, it is still not clear, from the architectural perspective, what configurations can lead to more robust DNNs. In this paper, we address this gap via a comprehensive investigation on the impact of network width and depth on the robustness of adversarially trained DNNs. Specifically, we make the following key observations: 1) more parameters (higher model capacity) does not necessarily help adversarial robustness; 2) reducing capacity at the last stage (the last group of blocks) of the network can actually improve adversarial robustness; and 3) under the same parameter budget, there exists an optimal architectural configuration for adversarial robustness. We also provide a theoretical analysis explaining why such network configuration can help robustness. These architectural insights can help design adversarially robust DNNs. Code is available at https://github.com/HanxunH/RobustWRN.

## 1 Introduction

Deep neural networks (DNNs) are becoming standard models for many real-world applications such as image classification [1], object detection [2] and natural language processing [3]. However, a line of research has shown that DNNs are vulnerable to adversarial examples (attacks), which can be easily crafted by slightly perturbing the input instance to maximize the model's prediction error [4–6]. This vulnerability of DNNs has become a major concern for their deployment in security-critical applications such as autonomous driving [7, 8] and medical diagnosis [9, 10].

A number of defense methods have been proposed to train adversarially robust DNNs [11–14], among which adversarial training has demonstrated the most promising results [15–17]. Adversarial training can be viewed as a type of data augmentation that trains DNNs on adversarial (instead of natural) examples [15–19]. Based on adversarial training, a set of works have been proposed to understand its learning and convergence behaviors, and the key factors for training adversarially robust DNNs. For example, it has been found that adversarial training encourages the model to learn more robust or compact features [20, 21], and it requires more data [22–25] or higher capacity models to gain more robustness [15, 26]. While these understandings have motivated several improved defense methods, it is still not clear, from an architectural perspective, *what makes an adversarially robust DNN*.

---

†Correspondence to: Xingjun Ma (danxjma@gmail.com)

35th Conference on Neural Information Processing Systems (NeurIPS 2021).

In this paper, we present the first comprehensive investigation on the architectural ingredients of adversarially robust DNNs. Our investigation is based on adversarial training and WideResNet-34-10 (WRN-34-10) [27], one extensively tested architecture in the defense literature. Based on the base architectural configuration of WRN-34-10, we apply a finely-controlled grid search to explore the impact of network width and depth configurations on the robustness of adversarial trained DNNs.

The standard WRN-34-10 consists of 3 stages with each stage being a group of 5 (i.e., depth) residual blocks and each residual block having 2 convolutional layers. We denote the three stages as Stage-1, Stage-2 and Stage-3 following the direction from the input to the output. Each stage is configured by a depth (number of residual blocks) and a width (number of filters) factor. The hyper-parameters for width and depth of each stage control the scale of learnable parameters (capacity). In this paper, we explore different configurations of width and depth for each of the three stages. Based on our explorations, we make the following key observations:

- Simply increasing the number of parameters (model capacity) by upscaling width or depth does not necessarily lead to improved robustness. This contrasts with current beliefs that, under the same type of architecture, more parameters (higher model capacity) can improve adversarial robustness [15, 26, 28]. Adversarial training does require larger capacity models, but there exists a trade-off. We provide both theoretical and empirical evidences that wider/deeper models increase Lipschitzness (larger Lipschitz constant).

- For a larger model used in adversarial training, reducing capacity at the last stage (Stage-3) of WRNs can achieve a better trade-off between capacity and Lipschitzness, thus improving adversarial robustness. This can be achieved by reducing either depth or width, with width reduction being slightly more effective. This highlights that smaller DNNs can also have better robustness if the parameter reduction is applied at the right place (i.e., the last stage).

- Under the same type of architectures (i.e., WRNs) and parameter budget, there may exist an optimal architectural configuration that can produce the most robust DNN. We show that the same configuration rule can also be applied to improve the robustness of VGGs, DenseNets (DNs), as well as networks found by Differentiable Architecture Search (DARTS).

Furthermore, we provide a series of understandings for the above findings, which can not only provide useful insights for training more robust models with adversarial training, but also shed new light on the architectural ingredients of adversarially robust DNNs.

## 2 Related Work

### 2.1 Adversarial Training

Adversarial training has been demonstrated to be the most reliable training method for obtaining adversarially robust DNNs [29, 30]. The standard adversarial training (SAT) can be formulated as a min-max optimization framework as follows:

$$\arg\min_{\boldsymbol{\theta}} \mathbb{E}_{(\boldsymbol{x},y)\sim\mathbb{D}}\Big[\max_{\boldsymbol{x}'} \mathcal{L}(f_{\boldsymbol{\theta}}, \boldsymbol{x}', y)\Big], \tag{1}$$

where the inner maximization generates adversarial examples $\boldsymbol{x}'$, the outer minimization trains the model on $\boldsymbol{x}'$, $f_{\boldsymbol{\theta}}$ denote the neural network and $\mathcal{L}(\cdot)$ is the cross entropy (CE) loss. During the inner maximization process, SAT uses PGD to generate adversarial examples [15]:

$$\boldsymbol{x}'_k = \Pi_{\epsilon}(\boldsymbol{x}'_{k-1} + \alpha \cdot \text{sign}(\nabla_{\boldsymbol{x}}\mathcal{L}(f_{\boldsymbol{\theta}}, \boldsymbol{x}'_{k-1}, y))), \tag{2}$$

where $\text{sign}(\cdot)$ is the sign function, $\boldsymbol{x}'_k$ is the adversarial example obtained at the $k$-th (for overall $K$ steps) perturbation step, $\alpha$ is the step size, and $\Pi_{\epsilon}$ is a projection (clipping) operation that projects the perturbation back onto the $\epsilon$-ball centered around $\boldsymbol{x}$ if it goes beyond.

Improved variants of SAT have also been proposed, such as the trade-off between adversarial robustness and natural accuracy (TRADES) [16], Dynamic AdveRsarial Training (DART) [17], Friendly Adversarial Training (FAT) [31], Misclassification Aware adveRsarial Training (MART) [18], Robust Self-Training (RST) [23], Unsupervised Adversarial Training (UAT) [32], Guided Adversarial Training (GAT) [33], Max-Margin AT [34], using Max-Mahalanobis Center (MMC) loss [35], accelerated AT [36–38], using pre-training [39], incorporating hypersphere embedding

[40], self-progressing robust training [41], Adversarial Weight Perturbation (AWP) [19], Adversarial Distributional Training (ADT) [42], Channel-wise Activation Suppressing (CAS) [21], Geometry-Aware Instance-Reweighted Adversarial Training (GAIRAT) [43] and robustness distillation [44, 45]. Adversarial Training has also been found to cause robust overfitting [46], but it can be mitigated by smoothing techniques [47].

## 2.2 Understanding Adversarially Trained DNNs

Understanding the working mechanism of adversarial training has been a hot research area. For example, it has been found that adversarial training encourages the model to learn more robust features [20, 48], have good generative ability [49, 50], improve the model's transferability to downstream tasks [51, 52] and improve performance on clean data [53]. It has also been found that using auxiliary training data with adversarial training can further improve adversarial robustness [22, 23], and that weight decay plays an important role in adversarial training [54]. Another important observation is that using WRNs instead of ResNets (RNs) can bring $\sim 3\%$-$5\%$ more robustness [16–18]. Other works also suggest that adversarial training requires deeper and wider models [15, 26, 55]. Also, the skip-connection operation used in WRN has been found can improve robustness for deeper architectures [56] and there exists a trade-off between depth and width for approximating natural functions [57]. On the other hand, there are also works showing that increasing the number of parameters for the same type of DNN architectures can only lead to limited robustness improvement [28, 58]; and wider networks may cause more perturbation instability [59].

Several recent works have applied neural architecture search (NAS) to search for more robust DNN architectures [60]. They found that, 1) densely connected cells result in improved robustness; and 2) under certain computational budget, adding convolution operations to direct connection edge is effective. Other works improve the NAS search strategy by searching on targeted capacity [61], maximizing certified lower bound [62], using the log-normal distribution to approximate the Lipschitz constant [63], using lower and upper confidence bounds in Bandit [64], or using perturbation-based regularization [65]. Another study on hand-crafted versus NAS-based architectures shows that, without adversarial training, NAS-based architectures are more robust for small-scale datasets and simple tasks than hand-crafted architectures, however, hand-crafted architectures are more robust than NAS-based architectures as the dataset size or the task complexity increases [66]. Note that NAS is extremely time-consuming, especially when applied with adversarial training. Previous works using NAS find optimal topological connections within the cell structure [60], but did not investigate depth/width configurations, which arguably has more impact on robustness (e.g., RNs vs. WRNs). In this work, we focus on fine-grained configuration exploration rather than blind search, which can produce more precise understandings of how depth and width affect robustness.

## 3 Wider and Deeper Models Increase Lipschitz Upper Bound

It has been theoretically shown that high Lipschitzness (larger Lipschitz constant) corresponds to low stability of the model's output to input perturbations [59]. However, adversarial training does require a larger capacity model (e.g., RN vs. WRN) [15], an empirical finding that goes against the theoretical expectation. In this section, we first theoretically show a trade-off between network capacity (width/depth) and the Lipschitz upper bound. In Section 4, we will empirically examine this trade-off and its relation to the improved adversarial robustness for larger capacity models.

The Lipschitz constant $L$ of a DNN measures the maximum rate of change in the output with the change in the input, and is closely related to adversarial robustness [4]. Formally, it is $\|f_{\boldsymbol{\theta}}(\boldsymbol{x}) - f_{\boldsymbol{\theta}}(\boldsymbol{x}')\| \leq L \|\boldsymbol{x} - \boldsymbol{x}'\|$.

**Theorem 1** (Lipschitz Constant Upper Bound of a Neural Network with Gaussian Distributed Weights)**.** *Consider an $n$ layer DNN $f$, where the weight parameters $\boldsymbol{\theta}$ are independent Gaussian random variables distributed as $\mathcal{N}(0, \sigma_{\boldsymbol{\theta}}^2)$ with $\sigma_{\boldsymbol{\theta}}^2$ denoting the variance of the Gaussian distribution, and where the activation functions are 1-Lipschitz. The expected Lipschitz constant of a DNN with hidden layer size $h$ is upper bounded by:*

$$L(f_{\boldsymbol{\theta}}) \leq \prod_{j=1}^{n} \left( \sqrt{h_{j-1}} + \sqrt{h_j} \right) \cdot \sigma_{\boldsymbol{\theta}_j}$$

**Theorem 2.** *For a convolutional neural network $f$, each layer's convolution operation with feature map size $W \times m \times m$ and kernel size $k \times k$, where the weight parameters $\boldsymbol{\theta}$ are independent Gaussian*

*random variables distributed as $\mathcal{N}(0, \sigma_{\boldsymbol{\theta}}^2)$, the expected Lipschitz constant is upper bounded by:*

$$L(f_{\boldsymbol{\theta}}) \leq \prod_{j=1}^{n} (m_j \sqrt{W_{j-1}} + (m_j - k_j + 1)\sqrt{W_j}) \cdot \sigma_{\boldsymbol{\theta}_j}$$

The proof for Theorem 1 and 2 is inspired by [67–69] and can be found in Appendix A.1. This establishes a connection between the upper bound on the Lipschitz constant of a feed-forward DNN with $n$ layers and width of $h_j$ for each layer. For the $j$-th layer of convolution operations, the Lipschitz constant upper bound increases with its input dimension ($W_{j-1} \times m_j \times m_j$) and the number of output channels $W_j$. More simply, it is upper-bounded by the variance of the weight matrix and the input representation's dimension plus the output representation's dimension. For the entire network, the Lipschitz constant upper bound grows exponentially with the depth. This suggests that wider and deeper models have a relatively larger change of the output due to the changes in the input, i.e., lower adversarial robustness.

Several works attempt to regularize the network's Lipschitz constant by using Parseval tight frames on the weight matrixes [70], enforcing constraints on the singular values of the weight matrixes [71], or via a Lipschitz-margin training [72]. However, a follow-up work points out that there exist both experimental and theoretical limitations for the above approaches [73]. Whilst the Lipschitz constant may not be used as a regularization, it has been widely adopted for analyzing the stability and adversarially robustness of DNNs [4, 59, 74].

## 4 Exploring Adversarially Robust Architectures

Our exploration of the relationship between DNN architectural configuration, Lipschitzness (size of the Lipschitz constant) and adversarial robustnessare starts with a fine-controlled grid search on the width/depth of the WideResNet (WRN) [27]. In Sections 4.1 and 4.2, we show our exploration results with depth and width, respectively. Based on these results, a pattern of robust depth/width configuration is discovered. In Section 4.4, we examine a linear scaling effect with the discovered robust configuration. In Section 4.5, we provide an analysis on the trade-off between model capacity and Lipschitzness, and the key factors contributing to improved adversarial robustness.

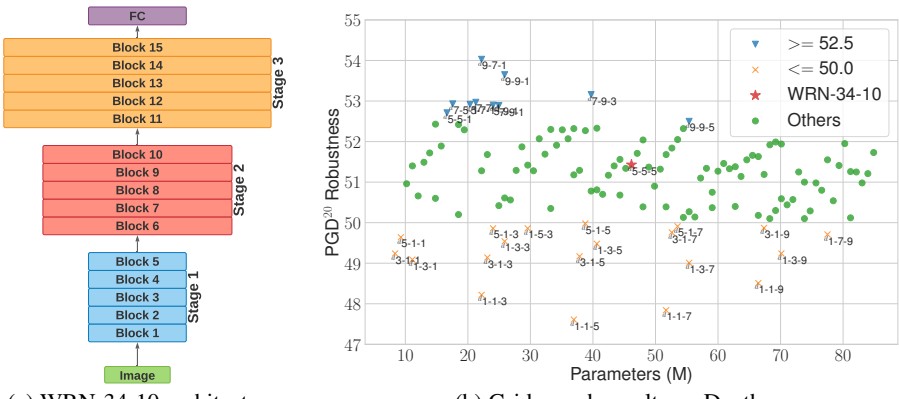

(a) WRN-34-10 architecture.   (b) Grid search results on Depth.

Figure 1: (a): Illustration of WRN-34-10 denoted as $^d$5-5-5. (b): Grid search results on different depth configurations. The three-digit numbers highlight the depth configurations of only those networks that have either a low ($<= 50.0\%$) or a high ($>= 52.5\%$) adversarial robustness (against PGD$^{20}$).

**Base architecture.** We take the standard WRN-34-10 designed for CIFAR-10 as our base architecture. Figure 1a provides an overview of the architecture and the detailed configurations are summarized in Appendix Table 3. The standard WRN architecture consists of 3 stages (groups) of residual blocks and 4 fixed convolutional layers. Here, we focus on the configuration of the 3 stages, which are the key components of the network. We denote the depth and width configuration for the $i$-th ($i \in \{1, 2, 3\}$) stage as $D_i$ and $W_i$, respectively. For standard WRN-34-10, $D_{1/2/3} = 5$ (denoted as $^d$5-5-5) and $W_{1/2/3} = 10$ (denoted as $^w$10-10-10). For the rest of this paper, we use $^dD_1$-$D_2$-$D_3$ and $^wW_1$-$W_2$-$W_3$ to represent the exact width and depth configurations. We explore the stage-wise depth and width configurations while keeping other configurations unchanged.

**Experimental settings.** We train all explored networks on CIFAR-10 dataset [1] using the standard adversarial training (SAT) with Projected Gradient Descent (PGD) [15] (see definition in equation (2)). Following the typical adversarial training setting, we constrain the $L_\infty$-norm of the maximum adversarial perturbation to $\epsilon = 8/255$, and use 10-step PGD (PGD$^{10}$) with step size $\alpha = 2/255$. After training, we test the robustness of the network on PGD adversarial examples crafted on the entire test set of CIFAR-10, under the same perturbation constraint $\epsilon = 8/255$. For evaluation, we use the 20-step PGD (PGD$^{20}$) with step size $\alpha = \epsilon/10$. The robustness is measured by the network's accuracy on the PGD$^{20}$ test adversarial examples. More details can be found in Appendix B.

## 4.1 Exploring Different Depths

We first explore different depth configurations based on the base WRN-34-10 architecture introduced above. For each of the three stages (e.g., Stage-1, Stage-2, Stage-3), we explore different depth $D_i \in \{1, 3, 5, 7, 9\}$. Since each stage has 5 possible depth configurations, the total number of all possible depth configuration for all 3 stages are 125 (5x5x5, permutation with replacement). We first perform a grid search on all the 125 depth configurations, then take a closer look at the impact at each individual stage. The adversarial robustness of the 125 networks (adversarially trained using SAT) against PGD$^{20}$ test adversarial examples is plotted in Figure 1b. Note that the depth configuration of standard WRN-34-10 is $^d$5-5-5. By investigating the robustness scores along the x-axis (number of parameters), we find that more parameters does not necessarily lead to improved robustness. For example, the networks with more than 80M (million) parameters are even less robust than some of those with only 20M parameters. Given the same level of parameters, for example $\sim$20M, different depth configurations can lead to $\sim 6\%$ difference in robustness. This implies that, *under the same parameter budget, there may exist an optimal depth configuration for adversarial robustness*.

Next, we take a closer look at the above grid search result and investigate the common characteristics of the top-5 most robust networks, the details of which are reported in Figure 2d. Interestingly, we find that the top-5 networks all have a significantly reduced depth of 1 or 3 at the last stage (i.e., Stage-3). This trend indicates that reducing model capacity at the last (deepest) stage can actually improve robustness. The other observation is that, having more residual blocks (higher depth) at the two shallow stages (i.e., Stage-1/2) can also improve robustness. For example, the top-2 networks have 9 residual blocks at Stage-1, and all top-4 networks have 9 or at least 7 residual blocks at Stage-2. This suggests that capacity is more important for the shallow layers. We conjecture this is because the network still needs sufficient capacity to learn the augmented examples by adversarial training. Note that the best performing model $^d$9-7-1 only uses half of the parameters of the standard WRN-34-10 ($^d$5-5-5), which is only ranked the 45-th out of all 125 models.

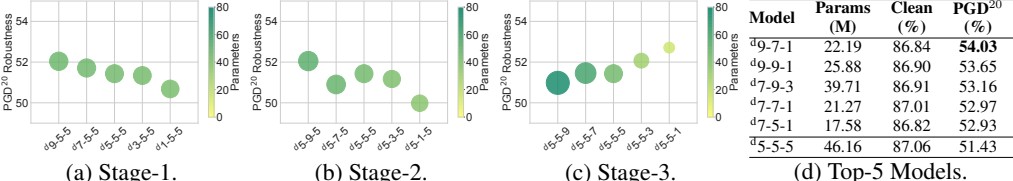

| | (a) Stage-1. | | | (b) Stage-2. | | | (c) Stage-3. | | | (d) Top-5 Models. | | | |

| Model | Params (M) | Clean (%) | PGD$^{20}$ (%) |
|---|---|---|---|
| $^d$9-7-1 | 22.19 | 86.84 | **54.03** |
| $^d$9-9-1 | 25.88 | 86.90 | 53.65 |
| $^d$7-9-3 | 39.71 | 86.91 | 53.16 |
| $^d$7-7-1 | 21.27 | 87.01 | 52.97 |
| $^d$7-5-1 | 17.58 | 86.82 | 52.93 |
| $^d$5-5-5 | 46.16 | 87.06 | 51.43 |

Figure 2: (a-c) The impact of depth on adversarial robustness at different stages. When studying one stage, the depths of other two stages are fixed to 5. (d) Clean accuracy and adversarial robustness of the top-5 most robust depth configurations discovered in the grid search. All networks are trained using SAT [15] on CIFAR-10. Robustness evaluated using PGD$^{20}$. $^d$5-5-5 is the depth configuration of the baseline WRN-34-10 model.

We further explore the distinctive impacts of depth on adversarial robustness at different stages via a control study. Specifically, we add or remove residual blocks from each individual stage of WRN-34-10 ($^d$5-5-5) while keeping the other two stages fixed to depth 5. The robustness results are shown in Figure 2a-2c. As can be observed, reducing depth at the first two stages constantly degrades the robustness, however, it is the other way around at the last stage (i.e., Stage-3). In relation to previous understanding that higher model capacity can lead to more robust models [28, 58], our finding indicates that it is true for the shallow layers but quite the opposite for the deeper layers. In other words, *more parameters can improve adversarial robustness only when added to the shallow layers* (e.g., layers in Stage-1 and Stage-2).

## 4.2 Exploring Different Widths

We further explore whether width also has a similar effect as depth. The standard WRN-34-10 has a width upscaling factor 10 applied to each stage, that is, $^{\text{w}}$10-10-10. Based on our above findings with the depth in Section 4.1, here we skip the grid search and directly investigate the impact of width at different stages. At each stage, we investigate different width configurations $W_i \in \{2, 4, 6, 8, 10\}$ for $i = 1, 2, 3$.

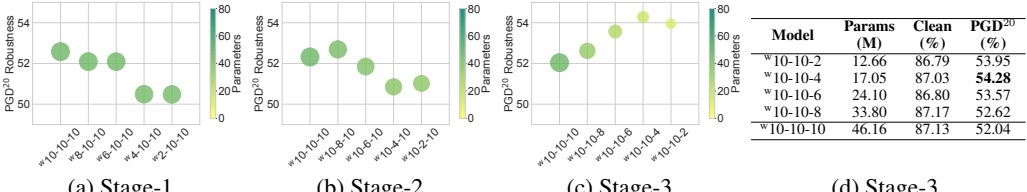

| Model | Params (M) | Clean (%) | PGD$^{20}$ (%) |
|---|---|---|---|
| $^{\text{w}}$10-10-2 | 12.66 | 86.79 | 53.95 |
| $^{\text{w}}$10-10-4 | 17.05 | 87.03 | **54.28** |
| $^{\text{w}}$10-10-6 | 24.10 | 86.80 | 53.57 |
| $^{\text{w}}$10-10-8 | 33.80 | 87.17 | 52.62 |
| $^{\text{w}}$10-10-10 | 46.16 | 87.13 | 52.04 |

(a) Stage-1.  (b) Stage-2.  (c) Stage-3.  (d) Stage-3.

Figure 3: (a-c): The impact of width on adversarial robustness at different stages. When studying one stage, the widths of other two stages are fixed to 10. (d): Clean accuracy and adversarial robustness of the networks obtained by reducing width in the last stage (i.e., Stage-3). All networks are trained using SAT [15] on CIFAR-10. Robustness is evaluated using PGD$^{20}$. $^{\text{w}}$10-10-10 is the depth configuration of the baseline WRN-34-10 model.

The robustness results are illustrated in Figure 3. We find that width reduction generally has a similar effect as depth reduction: reducing width at the first two stages harms robustness until $W_{1/2} = 4$, however, the same operation can improve robustness when applied to the last stage. This confirms the importance of high capacity at the shallow layers and low capacity at the deeper layers. Compared to depth reduction, we find that, with the same amount of robustness improvement, width reduction (at the last stage) can lead to smaller models. For example, $^{\text{w}}$10-10-4 (the second row in Figure 3d) achieves a similar robustness ($\sim$54%) as $^{\text{d}}$9-7-1 (the first row in Figure 2d). However, the number of parameters of the $^{\text{w}}$10-10-4 configuration is only 17.05M, which is much less than the 22.19M of the $^{\text{d}}$9-7-1 configuration.

Another interesting observation is that adversarial robustness does not change much if we reduce the width from $W_{1/2} = 4$ to $W_{1/2} = 2$ at Stage-1 or Stage-2, whereas the same reduction at Stage-3 hurts robustness. This is somewhat expected since, on one hand, the robustness might not be affected much unless a sufficient number of filters (channels) are removed, which is different to depth that configures the entire residual block. On the other hand, if too many filters are removed at the last stage, the network may lose the capacity required for proper learning, while in our depth exploration, there exists at least one residual block ($D_3 \geq 1$) at the last stage.

## 4.3 Exploring Depth-Width Combinations

Although reducing capacity at the last stage via either depth or width can improve robustness, there exists a limit. For example, if we reduce depth and width at the same time or too much of the width, the network may end up with insufficient capacity for proper learning. We first explore an extreme case that removes the entire Stage-3 as $^{\text{d}}$5-5-0. This ends up with 2% less robustness than baseline WRN-34-10. This result verifies the necessity of Stage-3. We then reduce depth and width simultaneously by setting the depth to $^{\text{d}}$5-5-1 and width to $^{\text{w}}$10-10-2. This produces a new network with a similar robustness ($\sim 52\%$) to PGD$^{20}$ as WRN-34-10. Note that, in this case, comparing to WRN-34-10, the number of parameters has been reduced by 70%. We then explore all the 25 possible depth-width combinations between the top-5 depth and width configurations in Figure 2d and 3d, respectively. Surprisingly, we find that none of these models can achieve better robustness than simply reducing the width to $^{\text{w}}$10-10-4. These models achieved the same level of robustness ($\sim 54\%$), but require more computations (FLOPS). For instance, the network with depth $^{\text{d}}$7-9-3 and width $^{\text{w}}$10-10-4 requires 2 times more FLOPS than WRN-34-10. This does not benefit adversarial training since it is known to be time-consuming. Although a more fine-grained (with decimals) exploration of the width configuration may lead to even more robust WRN models, here we simply take the $^{\text{w}}$10-10-4 configuration as our choice of the optimally-reduced WRN.

## 4.4 Scaling with the Discovered Configuration

Previous works [15, 16] have shown that a wider network like WRN-34-10 can be trained to be more robust than a standard ResNet like RN-34. Here, we investigate if we can obtain more robust models by scaling the discovered width configuration $^w$10-10-4. We test different scaling ratios $\gamma \in [0.25, 2.0]$, and show the robustness results in Table 1. Compared to $^w$10-10-4 ($\gamma = 1.0$), scaling down $\gamma$ to 0.5 or 0.25 decreases the robustness while scaling up $\gamma$ can further improve the robustness, although the improvement become less significant when $\gamma$ goes above 1.5. Note that the network

Table 1: Clean accuracy and adversarial robustness for scaled $^w$10-10-4 configurations. All networks are trained using SAT [15] on CIFAR-10 and evaluated using PGD$^{20}$.

| Scaling Ratio | Params (M) | Clean (%) | PGD$^{20}$ (%) |
|---|---|---|---|
| $\gamma = 0.25$ | 1.07 | 80.61 | 50.90 |
| $\gamma = 0.5$ | 4.27 | 84.31 | 54.07 |
| $\gamma = 1.0$ | 17.05 | 87.00 | 54.99 |
| $\gamma = 1.5$ | 38.33 | 87.68 | 55.33 |
| $\gamma = 2.0$ | 68.12 | 88.12 | **55.35** |

with $\gamma = 0.5$ has 10 times fewer parameters than the baseline WRN-34-10, but can already achieve a better robustness against PGD$^{20}$. The best robustness is achieved at $\gamma = 2.0$, i.e., $^w$20-20-8. We denote the corresponding WRN-34 network as WRN-34-R, more details can be found in Appendix Table 3. In Section 5, we will apply the $^w$10-10-4 configuration rule to more network architectures and evaluate their (along with WRN-34-R) adversarial robustness more systematically.

## 4.5 Empirical Understanding

We apply two closely related metrics, including Perturbation Stability [59] and Empirical Lipschitz [74] to explore the distinctive impact of the deeper layers to adversarial robustness. They measure the output stability of the neural network.

**Perturbation Stability.** Adversarial robustness is typically measured by the percentage of correctly classified adversarial examples, which can be further decomposed into the set of *correct clean* examples intersect with *stable* examples [59]. *Correct clean* examples refer to clean examples that can be correctly classified by the model $\{(\boldsymbol{x}, y) \sim \mathbb{D}, f_\theta(\boldsymbol{x}) = y\}$. *Stable* examples are defined as, $\{\boldsymbol{x} : \forall \boldsymbol{x}' \in \mathcal{X}, f_\theta(\boldsymbol{x}) = f_\theta(\boldsymbol{x}')\}$, where $\mathcal{X}$ is the domain of the $\epsilon$-ball around $\boldsymbol{x}$. This perturbation stability measures the fraction of examples whose outputs cannot be adversarially perturbed. While many factors can affect the model's performance on the *correct clean* examples such as the generalization capability of the model, the perturbation stability only measures if adversarial perturbations can change the prediction of the output. We apply this metric to understand the role of neural network architecture in adversarial robustness. More specifically we are interested to find out whether the improved robustness is a result of improved generalization, stability or both.

**Empirical Lipschitz constant.** The empirical Lipschitz constant is defined as [74]:

$$\frac{1}{n}\sum_{i=1}^{n}\max_{\boldsymbol{x}'\in\mathcal{X}}\frac{\|f_\theta(\boldsymbol{x}_i) - f_\theta(\boldsymbol{x}_i')\|_1}{\|\boldsymbol{x}_i - \boldsymbol{x}_i'\|_\infty}, \tag{3}$$

where $\mathcal{X}$ is the domain of the $\epsilon$-ball around $\boldsymbol{x}$ and $\boldsymbol{x}'$ can be generated by an adversarial attack (i.e., PGD$^{20}$). A lower value of the empirical Lipschitz constant implies a smoother and more adversarially robust classifier. For our analysis, we measure the empirical Lipschitz constant of the functions represented by the output layers of different residual blocks or the entire network (logits output). e.g., for block-5, we measure the maximum rate of change in its representation output between clean ($\boldsymbol{x}$) and adversarial ($\boldsymbol{x}'$) inputs.

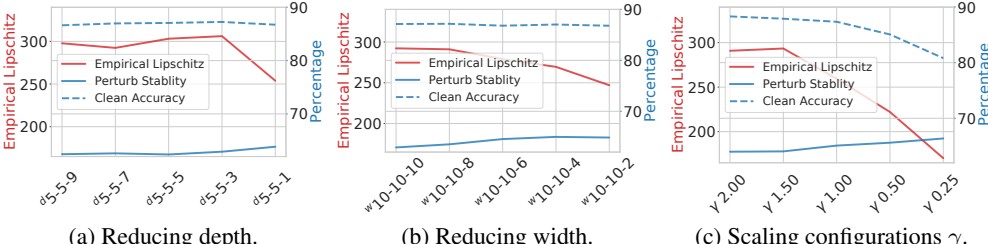

    (a) Reducing depth.        (b) Reducing width.        (c) Scaling configurations $\gamma$.

Figure 4: The change of perturbation stability and empirical Lipschitz constant when (a) depth of Stage-3 is reduced, (b) width of Stage-3 is reduced, or (c) linear scaling with $\gamma$ and $^w$10-10-4.

**The Trade-off Between Capacity and Lipschitzness.** We compute the above two metrics on the test set of CIFAR-10 for models with different depth and width configurations explored in Section 4.1

and 4.2. The results are illustrated in Figure 4, where the empirical Lipschitz constant is computed for the entire network. We can observe that, when depth or width for the given network is reduced, the empirical Lipschitz constant is also reduced, and the perturbation stability improves. This is consistent with our theoretical analysis in the Lipschitz constant upper bound. As shown in Figure 4b and 4c, this observation is more obvious for the width reduction. There exists a trade-off between the network capacity and Lipschitzness. For example, with $\gamma = 0.25$ scaling, the network achieves a much lower Lipschitzness and better stability, however, it also significantly reduces the clean accuracy. This indicates that adversarial training does require larger capacity models and a better trade-off can be achieved by balancing model capacity and Lipschitzness using proper architectural reduction.

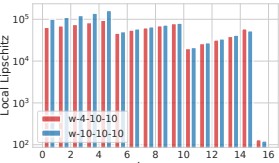 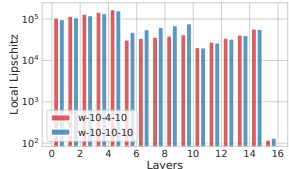 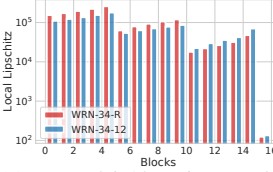

    (a) Reducing width at Stage-1.    (b) Reducing width at Stage-2.   (c) WRN-34-12 and WRN-34-R

Figure 5: Empirical Lipschitz constant of the output layers of different residual blocks (bins 1-15) or the entire network (bin 16). All experiments are run on CIFAR-10.

In Figure 5, we plotted the empirical Lipschitz constant of the output layer of each residual block or the entire network. The $f_{\boldsymbol{\theta}}(\boldsymbol{x})$ in equation (3) is replaced with the output of each residual block $f_{\boldsymbol{\theta}_j}(\boldsymbol{x})$ (from input to block output), and the last (16-th) bin is the empirical Lipschitz constant of the entire network (from input to logits). From Figure 5, we find that: 1) within each stage, the empirical Lipschitz increases with depth; 2) when transitioning from one stage to the next, the spatial dimension decreases while the empirical Lipschitz decreases; 3) comparing WRN-34-12 with WRN-34-R (Figure 5c), the empirical Lipschitz increase/decrease with the network width. This provides empirical results for our theoretical analysis in Section 3.

**Reducing Parameters at Deeper Layers Improves both Perturbation Stability and Lipschitzness.** Based on our theoretical analysis, reducing the width at Stage-1 and Stage-2 should improve the Lipschitzness of the corresponding stage as well as the entire network. However, empirically, it is true for the corresponding stage but not necessarily for the entire network. This is because the theoretical analysis in Section 3 only considers the interplay between two adjacent layers (or blocks), not including that of the non-adjacent layers. The empirical results in Figure 5 can fill this gap.

Specifically, Stage-1 width reduction (Figure 5a) lowers the Lipschitzness of Stage-1 blocks but not the overall Lipschitzness. Stage-2 width reduction (Figure 5b) can improve both Stage-2 and the overall Lipschitzness but fails to improve clean accuracy, perturbation stability nor adversarial robustness. WRN-34-R in Figure 5c marks our discovered reduction and scaling rule. Compared with standard WRNs, WRN-34-R not only reduces the width of Stage-3 (decreasing Lipschitzness) but also increases the widths of Stage-1 and Stage-2 (increasing Lipschitzness). Figure 5c shows that the increased Lipschitzness at Stage-1 and Stage-2 of WRN-34-R can be effectively mitigated at Stage-3, leading to decreased overall Lipschitzness and improved robustness (see Table 4.4). This also results in higher clean accuracy and perturbation stability (Figure 4c). We conjecture this is because Stage-3 (the last stage) is closer to the final output, thus has a more direct impact on the overall Lipschitzness. These empirical results provide a more in-depth understanding of the impact of width and depth configurations to the overall Lipschitzness, perturbation stability and adversarial robustness.

## 5 Adversarial Robustness Evaluation

In this section, we apply the discovered $^{\mathrm{w}}$10-10-4 width configuration rule to VGG, DenseNet (DN), DNNs discovered by NAS, WRN-34-R ($^{\mathrm{w}}$10-10-4 scaled by $\gamma = 2.0$), and evaluate their robustness with various adversarial attacks and defence methods on CIFAR [1] in the white-box setting. Additional results for CIFAR-10 black-box and ImageNet [75] using FastAT [37] can be found in Appendix D.

**Experimental Settings.** We consider VGG-11 [76], DenseNet-121 (DN-121) [77] and a network found by DARTS [78] with 11 cells. We denote the optimized VGG-11, DN-121 and DARTS networks as VGG-11-R, DN-121-R and DARTS-R (see Appendix B.2 for details), respectively. For a fair comparison between the discovered WRN-34-R (scaled by $\gamma = 2.0$) and the standard

WRN, we upscale WRN-34-10 to WRN-34-12 to make sure the two models have a similar amount of parameters. We train all networks using 4 adversarial training methods: Standard Adversarial Training (SAT) [15], TRADES [16], Misclassification Aware adveRsarial Training (MART) [18] and Robust Self-Training (RST) with 500K additional data [23]. More details are in Appendix B.1.

Table 2: White-box robustness results on CIFAR-10 and CIFAR-100. **500K:** Additional data used as in [23]. **Params:** number of parameters. SAT: Adversarial Training [15]; TRADES [16]; MART [18]; GAMA$^{100}$: 100-step GAMA attack [33]; AA: AutoAttack [30]; CW$_\infty$: $L_\infty$ version CW attack [79] optimized by PGD; **-R**: reconfigured networks following our discovered robust architectural configuration; **Last:** Results evaluated at the last checkpoint. **Best:** Results evaluated at the best checkpoint according to PGD$^{20}$. The best results are in **bold**.

| Dataset | Model | Method | Params (M) | Clean (%) | | FGSM (%) | | PGD$^{20}$ (%) | | GAMA$^{100}$ (%) | | CW$_\infty$ (%) | | AA (%) | |
|---|---|---|---|---|---|---|---|---|---|---|---|---|---|---|---|
| | | | | Last | Best | Last | Best | Last | Best | Last | Best | Last | Best | Last | Best |
| CIFAR-10 | VGG-11 | SAT | 9.23 | 79.24 | 77.84 | 55.98 | 56.68 | 42.62 | 45.46 | 38.59 | 40.65 | 45.45 | 46.29 | 37.21 | 39.84 |
| | VGG-11-R | SAT | 5.83 | 79.63 | 77.34 | **57.35** | **57.11** | 43.93 | 45.97 | 39.71 | 41.31 | 46.49 | 47.23 | 38.44 | 40.65 |
| | DN-121 | SAT | 6.96 | 86.87 | 86.07 | 65.56 | 66.58 | 51.67 | 54.79 | 48.60 | 51.00 | 52.03 | 54.00 | 47.16 | 50.34 |
| | DN-121-R | SAT | 6.00 | 87.22 | 86.01 | **67.12** | **67.20** | **52.52** | **55.16** | 49.37 | 51.44 | 53.07 | 54.67 | 47.75 | 50.54 |
| | DARTS | SAT | 6.58 | 86.76 | 86.55 | 64.48 | **67.10** | 49.44 | 54.23 | 46.52 | 50.74 | 52.03 | 54.00 | 45.16 | 49.98 |
| | DARTS-R | SAT | 2.53 | 87.20 | 85.79 | **66.74** | 66.61 | **52.36** | **55.01** | 48.71 | 50.94 | 53.07 | 54.67 | 47.75 | 50.54 |
| | WRN-34-12 | SAT | 66.46 | 86.71 | 87.20 | 64.06 | 66.26 | 49.92 | 53.09 | 47.45 | 50.40 | 52.23 | 53.58 | 46.06 | 49.18 |
| | WRN-34-R | SAT | 68.12 | 87.62 | 87.85 | **66.23** | **68.15** | **51.08** | **55.35** | 48.45 | 51.36 | 52.42 | 54.57 | 46.75 | 50.03 |
| | WRN-34-12 | TRADES | 66.46 | 85.84 | 84.59 | 65.70 | 66.85 | 53.02 | 56.01 | 49.60 | 52.35 | 53.35 | 54.72 | 48.48 | 51.83 |
| | WRN-34-R | TRADES | 68.12 | 86.77 | 86.02 | **67.99** | **68.49** | **55.15** | **57.66** | 51.92 | 53.86 | 55.41 | 56.30 | 50.90 | 53.46 |
| | WRN-34-12 | MART | 66.46 | 85.98 | 82.62 | 66.85 | 67.00 | 54.30 | 57.95 | 49.58 | 52.20 | 52.29 | 54.61 | 47.68 | 51.21 |
| | WRN-34-R | MART | 68.12 | 86.09 | 83.69 | **68.79** | **68.18** | **56.31** | **59.13** | 51.40 | 53.22 | 54.20 | 55.44 | 49.90 | 52.48 |
| CIFAR-10 +500K | WRN-34-12 | RST | 66.46 | 90.52 | 90.36 | 76.01 | 76.02 | 65.52 | 65.56 | 61.67 | 61.70 | 64.30 | 64.26 | 60.90 | 60.96 |
| | WRN-34-R | RST | 68.12 | 90.73 | 90.56 | **76.51** | **76.44** | **66.46** | **66.51** | 62.38 | 62.49 | 65.12 | 65.10 | 61.49 | 61.56 |
| CIFAR-100 | WRN-34-12 | SAT | 66.53 | 59.63 | 60.64 | 33.67 | 37.28 | 24.50 | 27.61 | 23.38 | 24.95 | 43.78 | 42.02 | 22.27 | 24.42 |
| | WRN-34-R | SAT | 68.16 | 61.17 | 61.33 | **35.00** | **38.72** | **25.03** | **29.02** | 23.38 | 25.70 | 43.52 | 41.46 | 22.72 | 25.20 |
| | WRN-34-12 | TRADES | 66.53 | 55.62 | 56.47 | 35.35 | 36.90 | 27.52 | 29.48 | 24.94 | 25.21 | 44.75 | **46.19** | 24.58 | 24.85 |
| | WRN-34-R | TRADES | 68.16 | 56.83 | 56.75 | **36.95** | **37.68** | **29.17** | **29.92** | 25.48 | 25.48 | 45.04 | 45.52 | 25.13 | 25.23 |
| | WRN-34-12 | MART | 66.53 | 58.51 | 57.29 | 36.06 | 39.48 | 26.50 | 32.43 | 23.85 | 27.64 | 41.53 | 38.73 | 23.33 | 26.92 |
| | WRN-34-R | MART | 68.16 | 61.72 | 58.27 | **39.68** | **41.24** | **29.94** | **34.12** | 26.27 | 29.33 | 39.20 | 38.45 | 25.60 | 28.63 |

**White-box Robustness.** We evaluate the robustness of the networks to 5 adversarial attacks including Fast Gradient Sign Method (FGSM) [5], Projected Gradient Descent (PGD) [15], Carlini and Wagner (CW) [79], Guided Adversarial Margin Attack (GAMA) [33] and AutoAttack (AA) [30]. We apply these attacks on the test sets of CIFAR-10 and CIFAR-100 with the same maximum adversarial perturbation $\epsilon = 8/255$ as adopted for model training. For PGD, we use the 20-step PGD (PGD$^{20}$) with step size $\alpha = \epsilon/10$. For GAMA attack, we set its perturbation steps to 100 following the original paper. We report the model's accuracy on the test adversarial examples crafted by these evaluation attacks for models obtained at both the best and the last checkpoints, following [16, 46, 18].

The white-box evaluation results including both clean accuracy and adversarial robustness are reported in Table 2. As can be inferred, a robustness gain of $1 \sim 3\%$ can be consistently achieved when the networks are reconfigured following our discovered configuration rule. And the improvements are not restricted to a particular architecture nor adversarial training method, except there is slight decrease for CW$_\infty$ on CIFAR-100 for SAT and MART. This wide range of robustness improvements by a simple architectural reconfiguration confirms that our findings are very general and can be immediately applied to commonly used DNNs to obtain more adversarial robustness. For VGG-11, DN-121 and DARTS, our robust reconfiguration can reduce the parameters by a considerable amount.

## 6   Relation to Existing Understandings

One recent work [59] shows that wider networks tend to increase the Lipschitzness, a finding that is consistent with ours. In [59], the theoretical analysis is based on Neural Tangent Kernel (NTK) under the assumption that all layers share the same width. By contrast, our analysis provides a more in-depth understanding related to the width and depth of each individual layer. Whilst in [59], the wider network utilizes a stronger regularization to mitigate the vulnerability (instability) caused by increased Lipschitzness, our work shows that this can be achieved alternatively by a simple reconfiguration of the architecture.

It has also been found that weight decay plays an important role in adversarial training [54, 80]. This can be explained by our theoretical analysis in Theorem 1 and 2. Considering the weight matrix is normally distributed $\mathcal{N}(0, \sigma_{\boldsymbol{\theta}}^2)$, weight decay encourages the model to learn weights of smaller magnitudes, thus reducing the variance $\sigma_{\boldsymbol{\theta}}^2$ of the weight matrix. This will lead to reduced upper bound of the Lipschitz constant and improved robustness. See Appendix E for more discussions.

# 7 Conclusion

In this paper, we explored the architectural ingredients of adversarially robust DNNs via extensive fine-controlled experiments and theoretical analysis. Our findings are: 1) more parameters does not necessarily lead to more adversarially robust models; 2) reducing capacity (up to a limit) via either depth or width at the deeper layers improves adversarial robustness; and 3) under the same type of architectures and parameter budget, there may exist an architectural configuration that can exploit the full robustness potential of the network. We also showed that depth and width offer different levels of flexibility for capacity reduction and robustness improvement. Following a width reduction and scaling rule, we showed that our findings are generic, not restricted to a particular adversarial training method, and can be immediately applied to improve both manually-designed or NAS-discovered DNNs. We also provide a series of empirical understandings on the distinctive impacts of the deeper layers on adversarial robustness. Our work can provide useful insights into the architectural perspective of adversarial robustness, and help design more adversarially robust DNNs.

## Border Impacts

Adversarial training is currently the most effective defense against adversarial attacks, although its performance is yet to be improved. In this work, we extensively studied the impact of network architecture to adversarial robustness. Our findings suggest that models can be made more robust by even reducing capacity at the deep layers. Such reduction can also help save the training cost of adversarial training which is known to be extremely time-consuming. We will open source the discovered architectural configurations to help future research design more robust architectures.

## Acknowledgment

Yisen Wang is partially supported by the National Natural Science Foundation of China under Grant 62006153, and Project 2020BD006 supported by PKU-Baidu Fund. This research was undertaken using the LIEF HPC-GPGPU Facility hosted at The University of Melbourne. This Facility was established with the assistance of LIEF Grant LE170100200.

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
