# A Proof for Theorem 1 and 2

## A.1 Theoretical Proof

The following is proof for Theorem 1 and 2 on Upper Bound on Lipschitz Constant of a DNN with Gaussian Distributed Weights, which is inspired by [67–69].

The Lipschitz constant upper bound of a $n$-layer DNN with 1-Lipschitz activation function (such as ReLU used in WRN) is:

$$L(f_{\boldsymbol{\theta}}) \leq \prod_{j=1}^{n} \|\boldsymbol{\theta}_j\|_2 \,, \tag{4}$$

where $\|\boldsymbol{\theta}_j\|_2$ is the spectral norms, i.e., the maximum singular values of the weight matrices $\boldsymbol{\theta}_j$.

**Theorem 3** (Gaussian Random Matrix [67]). *Let $\mathbf{A}$ be an $(N \times n)$ matrix whose elements are independent standard normal random variables. Then, $\sqrt{N} - \sqrt{n} \leq \mathbb{E}[\lambda_{min}(\mathbf{A})] \leq \mathbb{E}[\lambda_{max}(\mathbf{A})] \leq \sqrt{N} + \sqrt{n}$, where $\lambda_{min}$ and $\lambda_{max}$ denote the minimum and maximum singular values of $\mathbf{A}$, respectively, and $\mathbb{E}[\cdot]$ represents the expected value.*

Assuming each element in the weight matrix $\boldsymbol{\theta}$ follows normal distribution $\mathcal{N}(0, \sigma_{\boldsymbol{\theta}}^2)$, the expected maximum singular value of the $h_{j-1} \times h_j$ weight matrix $\boldsymbol{\theta}_j$ for layer $j$ is upper bounded:

$$\mathbb{E}[\|\boldsymbol{\theta}_j\|_2] = \mathbb{E}[\lambda_{\max}(\boldsymbol{\theta}_j)] \leq (\sqrt{h_{j-1}} + \sqrt{h_j}) \cdot \sigma_{\boldsymbol{\theta}}. \tag{5}$$

Combining equation (4) with equation (5), we have:

$$L(f_{\boldsymbol{\theta}}) \leq \prod_{j=1}^{n} (\sqrt{h_{j-1}} + \sqrt{h_j}) \cdot \sigma_{\boldsymbol{\theta}}. \tag{6}$$

This can be extended to convolutional neural networks (CNN). Using doubly block circulant matrix the convolution operation can be represented by matrix multiplication. Following [68], the convolutional operation can be rewritten as following:

$$\phi_i^{conv}(\vec{x}) = [V_{1,1} \ V_{1,2} \ ... \ V_{1,W_{j-1}}]\vec{x} + \vec{b}_i, \tag{7}$$

where the inputs and biases have been serialised into vectors $\vec{x}$ and $\vec{b}_i$, respectively, the $W_{j-1}$ is the number of channels (feature maps) of the previous layer. $W_j$ is known as the width of the convolution layer. The complete transformation constructed $W_j$ channels by adding additional rows to the block matrix:

$$\begin{bmatrix} V_{1,1} & \cdots & V_{1,W_{j-1}} \\ \vdots & \ddots & \\ V_{W_j,1} & & V_{W_j,W_{j-1}} \end{bmatrix} \tag{8}$$

where each matrix $V$ the doubly block circulant matrix with $m^2$ columns and $(m - k + 1)^2$ rows, where $m$ is the spatial dimensions (height and width) of the input representation (for simplicity we assume height and width are equal), the $k$ is the size of convolution kernel. Note, the doubly block circulant matrix is filled with kernel's weight ($\boldsymbol{\theta}$) and 0, which conform the assumption on normally distributed weights matrix $\mathcal{N}(0, \sigma_{\boldsymbol{\theta}}^2)$. Thus, for convolution neural networks, the Lipschitz constant upper bound is:

$$L(f_{\boldsymbol{\theta}}) \leq \prod_{j=1}^{n} (m_j \sqrt{W_{j-1}} + (m_j - k_j + 1)\sqrt{W_j}) \cdot \sigma_{\boldsymbol{\theta}}. \tag{9}$$

This can also be extended to residual blocks (used in ResNet). Following [68], the Lipschitz constant upper bound for the residual block with $n$ layers of convolution operation is:

$$L(f_{res}) \leq 1 + \prod_{j=1}^{n} \|\boldsymbol{\theta}_j\|_2 \tag{10}$$

Therefore, a ResNet consists of $n$ layers of residual blocks, the Lipschitz constant upper bound is:

$$L(f_{\boldsymbol{\theta}}) \leq n + \prod_{j=1}^{n} L(f_{res}) \tag{11}$$

### A.2 Empirical Verification on Gaussian Distribution Weights.

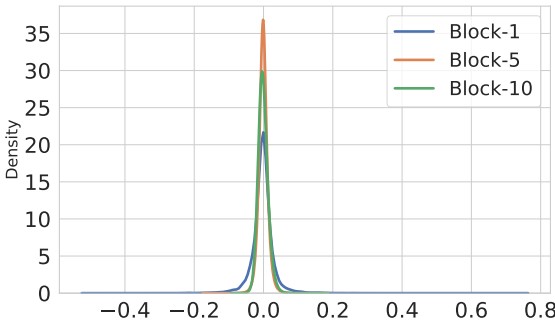

Figure 6: Kernel Density Estimation plot of the weight matrix for adversarially trained WRN-34-10. The weight matrix of the first convolution kenerl for block-1, block-5 and block-10.

## B  More Detailed Experiment Setup

### B.1  Training method

We train all networks (both the original and the optimized) using 4 adversarial training methods: Standard Adversarial Training (SAT) [15], TRADES [16], Misclassification Aware adveRsarial Training (MART) [18] and Robust Self-Training (RST) with 500K additional data [23]. For SAT, TRADES and MART, we apply their training strategies to train the networks for 100 epochs using Stochastic Gradient Descent (SGD) with initial learning rate 0.1, momentum 0.9 and weight decay $2 \times 10^{-4}$. The learning rate is divided by 10 at the 75-th and 90-th epochs. For RST, we set the weight decay to $5 \times 10^{-4}$, train for 400 epochs and use cosine learning rate scheduler [81] without restart. We train the networks on both CIFAR-10 and CIFAR-100 with maximum adversarial perturbation $\epsilon = 8/255$. For all training methods, we use the $PGD^{10}$ with step size $\alpha = 2/255$ for its inner maximization. All experiments are performed on NVIDIA Tesla P100 GPUs with PyTorch implementations. Code and pre-trained weights avaliable on Github https://github.com/HanxunH/RobustWRN.

### B.2  Network setup.

Table 3: Detailed configuration of the standard WRN. $D_i$ and $W_i$ denote the depth and width for the $i$-th stage, respectively. The total network depth is $\sum_{i=1}^{N=3} D_i$ plus 4 fixed layers. Within the same stage, the same type of residual blocks having 2 convolution operations are used. The final classification layer is omitted here. **WRN-34-10**: $D_{1/2/3} = 5$ and $W_{1/2/3} = 10$. **WRN-34-12**: $D_{1/2/3} = 5$ and $W_{1/2/3} = 12$. **WRN-34-R**: $D_{1/2/3} = 5$, $W_{1/2} = 20$ and $W_3 = 8$. Each residual block has 2 convolution layers with $3 \times 3$ kernels following the order of BN-ReLU-Conv (BN: batch normalization [82]; ReLU: ReLU activation; Conv: convolution).

| Group Name | Output Size | Block Details | |
|---|---|---|---|
| Conv-1 | 32×32 | [3×3, 16] | |
| Stage-1 | 32×32 | $3 \times 3, 16 \times W_1$ 
 $3 \times 3, 16 \times W_1$ | $\times D_1$ |
| Stage-2 | 16×16 | $3 \times 3, 32 \times W_2$ 
 $3 \times 3, 32 \times W_2$ | $\times D_2$ |
| Stage-3 | 8×8 | $3 \times 3, 64 \times W_3$ 
 $3 \times 3, 64 \times W_3$ | $\times D_3$ |
| Avg-Pool | 1×1 | [8×8] | |

We consider VGG-11 [76], DenseNet-121 (DN-121) [77] and a network found by DARTS [78] with 11 cells. VGG, DN and DARTS using similar stages design as WRN, VGG has 4 stages, DN and DARTS contain 3 stages. Following our discovered $^w$10-10-4 configuration, we reduce the 512 channels of VGG-11 and DN-121 of its last stage to 205 channels (i.e., $0.4 * 512$). The width configurations of each stages are [64, 128, 256, 205]. For DARTS, we use 11 cells and scale the width of the original set up by 2, the last stage reduced to 116 channels (i.e., $0.4 * 288$), the

width configuration is [108, 72, 144, 116]. We denote the optimized VGG-11, DN-121 and DARTS networks as VGG-11-R, DN-121-R and DARTS-R, respectively. For a fair comparison between the discovered WRN-34-R (scaled by $\gamma = 2.0$) and the standard WRN-34-10, we upscale WRN-34-10 to WRN-34-12 to make sure the two models have a similar amount of parameters.

## C  Empirical Understanding on Reduction in Stages 1 and 2.

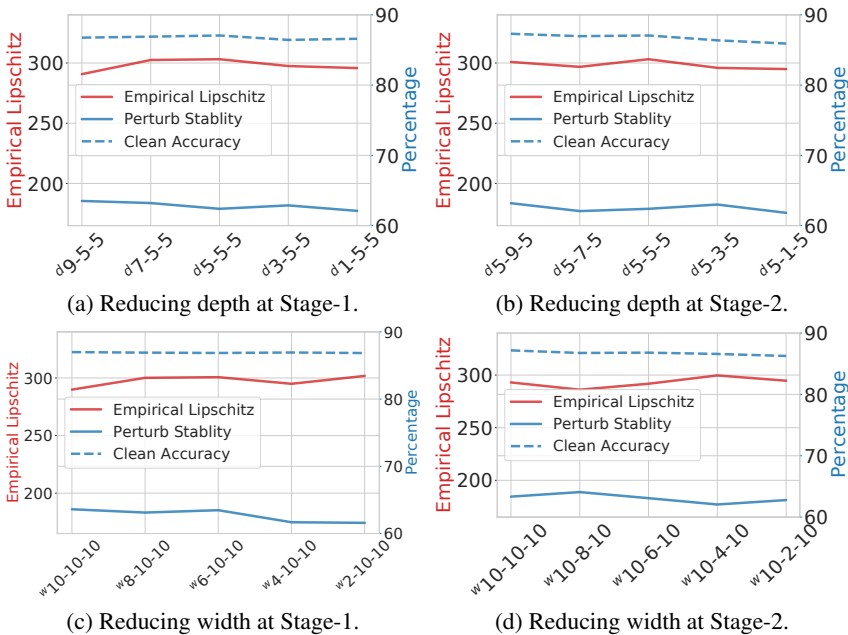

(a) Reducing depth at Stage-1.

(b) Reducing depth at Stage-2.

(c) Reducing width at Stage-1.

(d) Reducing width at Stage-2.

Figure 7: The change of perturbation stability and empirical Lipschitz constant when (a) depth of Stage-1 is reduced, (b) depth of Stage-2 is reduced (c) width of Stage-1 is reduced, or (d) width of Stage-2 is reduced.

We apply the same analysis as in Section 4.5 for reducing depth or width in shallower stages (1 and 2). Following the same procedure as in Figure 4, we plot the results for reducing the capacity using width and depth for shallower stages. As shown in Figure 7, both clean accuracy and perturbation stability decrease as width or depth is reduced. Therefore, it decreases the overall adversarial robustness. This result highlighted that the trade-off between the capacity and Lipschitzness can only be effectively mitigated by reducing the capacity of the last stage.

## D  Additional Robustness Evaluation

### D.1  Black-box Robustness

We explore whether the robustness improvements are still valid in a black-box setting. Following a standard black-box robustness evaluation setting [15, 16], here we apply FGSM, PGD[20] and $CW_\infty$ attacks on a naturally-trained surrogate model to craft test adversarial examples, then test the robustness of adversarially-trained WRN-34-12 and WRN-34-R models on these adversarial examples. We use ResNet-50 for the surrogate model, and train WRN-34-12 and WRN-34-R using SAT, TRADES and MART. This experiment is conducted on CIFAR-10, and the results are reported in Table 4.

Similar to the white-box results, the discovered robust reconfiguration can consistently improve the black-box robustness of the networks, regardless of the methods used for adversarial training. This verifies that the robustness can indeed be improved by simple architectural reconfiguration, either white-box or black-box. Although there is still much room for improvement, we believe our findings are useful for the community to better understand what type of architectural configurations can help adversarial robustness.

Table 4: Black-box robustness results on CIFAR-10. A naturally-trained ResNet-50 surrogate model is used to craft the black-box (transferred) attacks. The best results are in highlighted **bold**.

| Model | Method | Clean (%) | FGSM (%) | $\text{PGD}^{20}$ (%) | $\text{CW}_\infty$ (%) |
|---|---|---|---|---|---|
| WRN-34-12 | SAT | 87.09 | 85.03 | 85.15 | 86.30 |
| WRN-34-R | SAT | 88.04 | **85.83** | **86.10** | **87.40** |
| WRN-34-12 | TRADES | 84.67 | 82.83 | 82.97 | 83.80 |
| WRN-34-R | TRADES | 85.36 | **83.28** | **83.40** | **84.71** |
| WRN-34-12 | MART | 82.94 | 80.66 | 80.81 | 81.80 |
| WRN-34-R | MART | 83.75 | **81.56** | **81.66** | **82.71** |

## D.2 ImageNet

For ImageNet, we trained the models using FastAT [37], and followed its training/testing setting. We used the code from the public available GitHub repository[*]. We followed the $\epsilon = 4/255$ and PGD adversary using 10 steps. The results are reported in Table 5. We reproduced the result for ResNet-50. For ResNet-50-R, we applied our discovered configuration, i.e., reducing the width of the last stage to 40% and scale-up the entire model to have the same amount of parameters as ResNet-50.

Table 5: Robustness ($\epsilon = 4/255$) results for ResNet-50 on ImageNet.

| Dataset | Model | Clean (%) | $\text{PGD}^{10}$ (%) |
|---|---|---|---|
| ImageNet | ResNet-50 | 55.45 | 30.48 |
| ImageNet | ResNet-50-R | 56.63 | **31.14** |

# E Additional Discussion

## E.1 Performance of CW attack on CIFAR-100

On CIFAR-100, the robustness for discovered configurations is not as good as the baseline model. This could be due to the $\text{CW}_\infty$ attack we used in Table 2 is the weaker version that has been commonly used as a more efficient alternative to its original version for robustness evaluation. The margin-based attacks may suffer from the imbalanced gradients problem on some defence models, as revealed in a recent work [83]. In comparison, AutoAttack (AA) is stronger and more reliable than other attacks as a robustness evaluation. The discovered architectural reconfiguration demonstrates consistent improvement across multiple datasets, DNN architectures, and adversarial training methods as shown in Table 2.

## E.2 Auto-Attack leaderboards

Table 6: Auto-Attack leaderboards. Results are reported base on Auto-Attack's GitHub Page with models using additional data as in RST [23].

| Venue/Year | Method/Paper | Model | AutoAttack(%) |
|---|---|---|---|
| Arxiv-2020 | Gowal *et al.* [80] | WRN-70-16 | 65.88 |
| NeurIPS-2021 | **Ours+EMA** | WRN-34-R | 62.54 |
| NeurIPS-2021 | **Ours** | WRN-34-R | 61.56 |
| NeurIPS-2021 | Wu *et al.* [59] | WRN-34-15 | 60.65 |
| NeurIPS-2020 | AWP [19] | WRN-28-10 | 60.04 |
| NeurIPS-2019 | RST [23] | WRN-28-10 | 59.53 |
| NeurIPS-2020 | Hydra [84] | WRN-28-10 | 57.14 |
| ICLR-2020 | MART [18] | WRN-28-10 | 56.29 |

The AutoAttack [30] is an ensemble attack method that is currently the most reliable and widely acknowledged evaluation benchmark in Adversarial Defences. According to the leaderboards[†], there is significant use of ResNet/WRN with increasing model complexities (SOTA method uses WRN-70-16). Our theoretical and experimental results show that there exists a trade-off between Lipschitz constant upper bound and the model complexity. This provides a different insight into how

---

[*]FastAT GitHub

[†]AutoAttack GitHub

the architectures of DNN affect adversarial robustness. Moreover, adversarial defence research is now at the stage where even 1%-2% improvement on AA is significant enough to propose a new defence method. Current SOTA defences use larger models [59, 80], where WRN-34-R has similarly amount parameters with WRN-34-12, that can achieve 1% improvement over WRN-34-15 using larger regularization strength [59]. In addition, [80] explored tricks in adversarial training, such as reproduce additional data, weight averaging, and change activation functions.

Our results in Table 2 follows the original settings and hyperparameters described in the corresponding papers. Here, we test if our discovered model can gain further robustness by incorporating the exponential moving average (EMA) [80] which is adapted from Stochastic Weight Averaging [85]. Results show that with EMA, our discovered configuration for WRN can further gain additional 1% robustness on AA. This further demonstrated that this configuration can consistently improve robustness on a wide range of adversarial training methods.