# OpenReview forum: "Exploring Architectural Ingredients of Adversarially Robust Deep Neural Networks"
_NeurIPS.cc/2021/Conference — NeurIPS 2021 Poster_

### Official Review · Reviewer_55GK · 2021-06-29

**Rating:** 7
**Confidence:** 4

**Summary:**

In this paper, the authors evaluate the impact of network width and depth on the robustness of models trained through adversarial training.  Main contributions:
1. Theoretically show that increasing network width and depth leads to lower perturbation stability
2. Empirically evaluates the impact of changing depth and width and various stages of the WRN-34-10 architecture.  The authors find that decreasing the depth and width at the last stage can lead to higher robustness while decreasing depth and width at earlier stages of the network hurts robustness.
3. Find a width configuration that has high robustness on WRN-34-10 and show that this configuration can also be used to improve robustness on a variety of other base architectures.  The authors show that this pattern holds for various datasets and adversarial training techniques.

**Limitations And Societal Impact:**

See main review for comments.  My main concern is that in Section 5, which empirically shows that the architecture configuration found by the authors generalizes across models/datasets/training techniques, only focuses on varying width while previous sections also analyzed the impact of depth.  These patterns in depth should ideally generalize as well, but there are no experiments to show this.

**Main Review:**

Originality: To the best of my knowledge, the design of the experiments is novel.  While previous works have looked into the impact of width and depth [1,2], in those experiments, the authors vary the overall width and depth of the network.  The authors of this paper perform a more in depth experiment in which they modify the width and depth of stages within the model, which allows for a better understanding of where in the architecture more width and depth helps.  On the theoretical result of how Lipschitz constant is upper bounded by some function of width and depth, [2] also provides an upper bound on Lipschitz constant related to network width and depth, so a few sentences comparison between the theoretical results from this paper and [2] should be added.

Quality: There is theory to show that width and depth decreases stability and there is good scope in experiments to show that decreasing width and depth at the last stage of the network improves robustness while earlier stages perform better with higher width and depth.  Additionally, experiments are performed along various architectures, datasets, and adversarial training methods to show that their findings generalize.  In terms of scope of experiments, I do have a few suggestions/questions:
- in section 4.3, the scaling is performed on the discovered width configuration.  What about the impact on scaling for depth?  Additionally, experiments in section 5 use the -R models which is the scaled width configuration.  What about varying the depth configuration for these models?  Ideally, we should see both the depth and width trends found in WRN-34-10 to hold for these other models as well.
- in section 4.4, Figure 4 includes plots of empirical Lipschitz constant and perturbation stability for varying width and depth at only the last layer.  I think it would be helpful to include plots for varying width and depth at earlier layers as well especially since Figure 5 suggests that reducing width at earlier layers does not necessarily reduce the Lipschitzness of the network.  Potentially, reducing depth and width at earlier stages would just decrease the clean accuracy while having little impact on the perturbation stability and empirical Lipschitz which I think ties in with the hypothesis that more parameters are needed in the earlier stages for the model to fit to the adversarial images during training.

Clarity: Overall, the writing is clear.  I feel like the transition between section 3 and 4 is a little awkward though since the first 3 subsections of Section 4 are unrelated to the theory.  A potential solution is to group 4.1-4.3 in 1 section and section 3 and 4.4 within another section.  Some minor comments:
- in Section 3, it would help to add a reference to Appendix A.2 to show that the Gaussian distributed weights assumption holds for an adversarially trained model.
- line 149, missing space between the words "robustness" and "are"
- the trends in Figure 5 are a little difficult to see

Significance: The results help guide researchers on how to design architectures that improve adversarial robustness.  The authors indicate that robustness can be improved even when decreasing the number of parameters by reducing the width/depth at the last stage of the network, which is useful in situations with limited storage.

[1] Gowal, Sven, et al. "Uncovering the Limits of Adversarial Training against Norm-Bounded Adversarial Examples." arXiv preprint arXiv:2010.03593 (2020).
[2] Wu, Boxi, et al. "Does Network Width Really Help Adversarial Robustness?." arXiv preprint arXiv:2010.01279 (2020).

**Time Spent Reviewing:**

3

---

> ### Author Response · Authors · 2021-08-10
> **Response to Reviewer 55GK**
>
> Thanks for your insightful comments and useful suggestions. Please find our responses below to your questions.
>
> ---
> **Q1:** Comparison between the theoretical results from this paper and [2] should be added.
>
> **A1:**
> The comparison and relationship of our work to [2] has been discussed in Section 6 “Relation to Existing Understandings” (Lines 343 - 349). [2] studied the impact of network width on perturbation stability and adversarial robustness, while our analysis covers both width and depth and the finding on width is consistent with [2]. Our theoretical analysis provides more fine-grained results into different layers and stages, while [2] focuses on the width of the entire network. From the empirical side, [2] exploit their finding in a form of regularization to improve robustness, while we simply reconfigured the deeper layers of a network based on the discovered reduction and scaling rule. This allows us to improve a decent amount of adversarial robustness for a wide range of models with fewer (or similar if scaling rule is applied) model parameters (Table 6 in Appendix E.1). We will add more comparisons to Section 6.
> ---
>
> **Q2:** Scaling for depth in Section 4.3
>
> **A2:** Depth scaling is slightly more difficult to operate than width scaling, i.e. the computational cost (FLOPS) is higher than scaling with width. While we would expect the same trend to hold for depth scaling, as investigated in [1, 3] for WRN as well as in section 4.3 for “-R” models. We will add an empirical analysis of the suggested scaling to the revision.
>
> ---
> **Q3**: Figure 4, varying width and depth at **early** stages 1-2.
>
> **A3**: Thanks for the suggestion and the insightful observation. We have plotted new plots with varied width and depth at earlier stages and investigated the changes in **Lipschitzness of the entire network** and the **clean accuracy**. We found that varying width and depth at the earlier layers does not significantly impact  the Lipschitzness, while the clean accuracy decreases as width/depth is reduced, just as you expected. Since we are unable to show the plots here, we will add them to the revision. We thank the reviewer again for this helpful discussion; it will make our analysis “The Trade-off Between Capacity and Lipschitzness” in Section 4.4 more complete.
>
> ---
>
> [1] Gowal, Sven, et al. "Uncovering the Limits of Adversarial Training against Norm-Bounded Adversarial Examples." arXiv preprint arXiv:2010.03593 (2020).\
> [2] Wu, Boxi, et al. "Does Network Width Really Help Adversarial Robustness?." arXiv preprint arXiv:2010.01279 (2020).\
> [3] Madry, Aleksander, et al. Towards deep learning models resistant to adversarial attacks. ICLR 2018.

---

> > ### Comment · Reviewer_55GK · 2021-08-14
> > **Clarification about depth experiments**
> >
> > Thank you for the response.  I think most of my points have been addressed except for the depth experiments.  More importantly than depth scaling, my main concern for Q2 was that the -R models only combine the results found for width scaling and the width configuration found.  Why not also use the 9-7-1 depth configuration from section 4.1 which seems to have the best performance combined with the -R width configuration?  I am curious to know if combining the results found in the depth study with the width study can lead to even better performance in section 5.

---

> > > ### Author Response · Authors · 2021-08-15
> > > **Response to the depth experiments**
> > >
> > > Thank you very much for the feedback.
> > >
> > > You can find our analysis of combining depth and width reductions in Appendix C.
> > > We explored all the 25 possible depth-width combinations between the top-5 depth configurations in Figure 2d and the top-5 width configurations in Figure 3d, and found that none of these combinations can achieve better robustness than simply reducing the width to $^{W}$10-10-4.
> > >
> > > The 1st table below reports the detailed results of fixing depth reduction to optimal ($d$-9-7-1 ) while varying the width reduction. The 2nd table shows the robustness is still not the best even after an upscaling.
> > >
> > > While reducing both depth and width can improve Lipschitzness and stability, it also tends to harm the clean accuracy which in turn reduces the robustness. This phenomenon can be interpreted as an architectural perspective *robustness-accuracy* tradeoff [1]. Please also find our *Capacity-Lipschitzness* tradeoff analysis at Lines 275-285 “The Trade-off Between Capacity and Lipschitzness.”
> > >
> > >
> > > [1] Robustness May Be at Odds with Accuracy. ICLR 2019.
> > >
> > >
> > > ---
> > > `Table 1: Fixing depth reduction to $d$-9-7-1 while varying width reduction on CIFAR-10.`
> > >
> > > |  Width-Factors  	| Clean 	| PGD$^{20}$ 	| Stability 	|   Lip  	|
> > > |:---------------:	|:-----:	|:----------:	|:---------:	|:------:	|
> > > |  $^{W}$-10-10-2 	| 86.29 	|    53.90   	|   65.47   	| 208.30 	|
> > > |  $^{W}$-10-10-4 	| 86.28 	|    53.70   	|   65.24   	| 215.10 	|
> > > |  $^{W}$-10-10-6 	| 86.72 	|    53.60   	|   64.70   	| 209.82 	|
> > > |  $^{W}$-10-10-8 	| 86.76 	|    53.60   	|   64.75   	| 219.16 	|
> > > | $^{W}$-10-10-10 	| 86.48 	|    53.51   	|   64.85   	| 214.30 	|
> > >
> > > ---
> > >
> > > `Table 2: Row 1: the performance of $d$-9-7-1; Row 2: applying width reduction to $d$-9-7-1; Row 3: upscaling the model in the second row by a factor of $\gamma=2.0$. `
> > >
> > > |         Model        	| Clean 	| PGD$^{20}$ 	|
> > > |:--------------------:	|:-----:	|:----------:	|
> > > |      $^{d}$9-7-1     	| 86.48 	|    53.51   	|
> > > |   + Width Reduction  	| 86.28 	|    53.70   	|
> > > | + Scale $\gamma=2.0$ 	| 87.04 	|    52.74   	|
> > > ---

---

> > > > ### Comment · Reviewer_55GK · 2021-08-15
> > > > **Suggestion about writing**
> > > >
> > > > I believe that this response addresses my concern about depth experiments.  One suggestion I have about the writing is adding a few sentences to the beginning of Section 5 to make it clear that the -R model configuration tested is actually the best configuration found over width and depth combinations.  This would help improve the transition to this section since previous sections test both depth and width configurations while the -R model tested in Section 5 is only a width configuration.
> > > >
> > > > I have read the concerns of the other reviewers and authors responses and believe that these concerns have been addressed.  Since my main concern was due to my confusion over why Section 5 only tests a width configuration and has been addressed, I am raising my score to a 7.

---

> > > > > ### Author Response · Authors · 2021-08-16
> > > > > **Will revise as suggested**
> > > > >
> > > > > Thanks for the advice. We will make sure Section 5 is properly revised as suggested.

---

### Official Review · Reviewer_hME6 · 2021-07-16

**Rating:** 6
**Confidence:** 3

**Summary:**

This paper conducts an empirical study on the effect of the model architectural on the model robustness. The experimental results show that more parameters do not necessarily lead to more robust model. Reduce capacity of the deeper layers can improve the robustness. With the same architectures, there may exist one optimal architectural configuration for producing the most robust DNN.

**Limitations And Societal Impact:**

The societal impact is well discussed.

**Main Review:**

# Originality
The results of this paper are new and interesting. Readers can learn a lot of new knowledge from the empirical study. The related work is also well discussed.

# Quality
Overall, the paper is clear and the claims can be supported well by the results. Some detailed comments:
1. The study mainly focuses on WRN and CIFAR-10. It is not clear how can the findings generalise to other models or datasets. I appreciated the effort of the authors in applying the similar rules into other datasets (e.g., ImageNet). However, it will be more convincing if more results and analysis can be provided.
2. In section 5, the paper said that they applied the discovered configuration rule on the new models. It is not clear what are these rules and how to conduct such rules in the different models. For example, reduce capacity of deeper layers?  which layers are reduced?
3. The paper claimed that reducing capacity of the deeper layers can improve the robustness. Why? More theoretical analysis would be helpful.
4. In Table 2, it shows that the results from $CW_\infty $ on CIFAR-100 are not as what you expect. Could you provide more analysis?

# Clarity
The paper is written clearly and easy to follow.

# Significance
The results are promising and insightful.  Other researchers can benefit from the results

**Time Spent Reviewing:**

5 hours

---

> ### Author Response · Authors · 2021-08-10
> **Response to Reviewer hME6**
>
> Thanks for your valuable comments. Please find our responses below for your questions.
>
> ---
> **Q1:** More results and analysis.
>
> **A1:** The reason why we use the WRN model and CIFAR-10 dataset for the exploration is simply that it is the most extensively studied setting in adversarial defense. We have tested the discovered scaling rule on most of the commonly used models, datasets, and adversarial training methods in the field. In response to the comment, we have run an additional experiment with MobileNet-v2 [1]. We applied both width reduction and scaling (more details in A2). Due to the design of MobileNet-v2, for simplicity, we use the width factor to describe our configuration. The width factor is the notation $t$ used in the MobileNet-v2 paper [1], and it is the same concept for the width factor for WRN. The width factors for the original configuration are [6, 6, 6, 6] for all stages (four stages in total). Applying our reduction rule, we reduce the width factor of the last stage to 2.4 then upscale the entire network by 2 ([12, 12, 12, 4.8]) for each stage, denoted as MobileNet-v2-R. Following the same experimental setting in Table 2 (**Last:** results evaluated at the last checkpoint. **Best:** results evaluated at the best checkpoint according to PGD$^{20}$.), the white-box robustness results can be found in the table below, where it demonstrates consistent robustness improvements. We are happy to run more experiments that are of the reviewer’s particular interest.
>
> |     Model     	| Method 	| Params 	| Clean Last 	| Clean Best 	| PGD$^{20}$ Last 	| PGD$^{20}$ Best 	| AA Last 	| AA Best 	|
> |:-------------:	|:------:	|:------:	|:----------:	|:----------:	|:-------------:	|:-------------:	|:-------:	|:-------:	|
> |  MobileNet-v2  	|   SAT  	|  2.30  	|    82.77   	|    81.36   	|     51.24     	|     51.59     	|  45.77  	|  45.33  	|
> | MobileNet-v2-R 	|   SAT  	|  2.58  	|    83.88   	|    82.29   	|     **51.39**     	|     **52.65**     	|  **46.12**  	|  **47.26**  	|
>
> ---
>
>
> **Q2:** Discovered pattern and rules in Section 5
>
> **A2:** We discovered two rules: 1) a reduction rule and 2) a scaling rule. The reduction rule defines how much (e.g., 60%) the width should be reduced and where (e.g., the last stage of a ResNet). The scaling rule defines how much the width of the entire network should be upscaled to gain additional robustness (similar to ResNet -> WideResNet). In Section 5 for VGG/DenseNet/DARTS, we applied the reduction rule and ignored the scaling rule. We will add scaling experiments for these models to the revision.
>
> Most modern DNN architectures consist of multiple stages of stacked DNN layers (e.g., both VGG and DenseNet have 4 stages), and as illustrated in Figure 1a), the stage connected to the output is denoted as the last stage. For VGG-11/DenseNet-121, the original width for each stage are [128, 256, 512, 512] and [64, 128, 256, 512] respectively. Applying the reduction rule, the widths of their last stages are both reduced from 512 to 205. Their reduced versions are denoted as VGG-11-**R** and DN-121-**R** respectively. Other architecture configurations are kept the same as their original versions. For DARTS, we used one simplified version of 11 stacked cells but 2 times wider and reduced the width of its last stage by 60%. The widths for each stage are [72, 144, 288] and [72, 144, 116] for DARTS and DARTS-R. We will make this clearer in the revision.
>
> ---
> **Q3:** Why can reducing the capacity of the deeper layers improve the robustness?
>
> **A3:** We leverage the theory of Lipschitzness and its associated perturbation stability to adversarial robustness. For robustness, our finding reveals that reducing the parameter redundancy of the deeper layers can lead to more adversarially robust models. Relating to the robust vs. non-robust features explanation [2], our results indicate that redundant parameters at the deep layers can cause the network to learn more non-robust features, meaning that non-robustness features are more associated with the complex correlations represented at the deeper layers, while robust features are more associated with shallow layers. This also explains why adversarial training needs the shallow layers to be as wide as possible, as it needs to learn much more robust features.  We hope this new angle of explanation can help address your question. We will add empirical analysis to the revision.
>
> ---
>
> **Q4:** Performance of CW attack on CIFAR100
>
> **A4:** The CW attack we used here is the margin loss defined CW$_\infty$, a weaker version that has been commonly used in the literature as a more efficient alternative to its original version for robustness evaluation. The margin-based attacks may suffer from the imbalanced gradients problem on some defense models, as revealed in a recent work [3]. Please note that the AutoAttack (AA) is stronger and more reliable than other attacks, and has become the new standard attack for robustness evaluation.  According to AA, the discovered architectural reconfiguration demonstrates consistent improvement across multiple datasets, DNN architectures, and adversarial training methods (Table 2 and Table 6 in appendix E.2).
>
> ---
> [1] MobileNetV2: Inverted Residuals and Linear Bottlenecks. CVPR 2018.\
> [2] Adversarial Examples Are Not Bugs, They Are Features, NeurIPS 2019.\
> [3] Imbalanced Gradients: A New Cause of Overestimated Adversarial Robustness. arxiv, 2021.

---

> > ### Comment · Reviewer_hME6 · 2021-08-31
> > **The response has addressed my concerns.**
> >
> > Thanks for your response. I think most of my concerns have been addressed. I suggest the authors to add these explanation in the paper for better understanding.

---

> > > ### Author Response · Authors · 2021-08-31
> > > **Thanks for the suggestion**
> > >
> > > Thank you very much for the positive feedback and suggestion. We will make sure the explanations are added to the revision.

---

### Official Review · Reviewer_DNQQ · 2021-07-17

**Rating:** 5
**Confidence:** 4

**Summary:**

The authors attempt to relate adversarial robustness with architectural properties of neural networks such as width and depth. They also provide some theoretical analysis in terms of Lipshitz upper bounds. There are some interesting empirical observations, but no definite conclusions can be drawn.


**Limitations And Societal Impact:**

Yes

**Main Review:**

- There are attempts to explain network robustness with Lipschitz upper bound, such as the theoretical results derived in section 3. But the bounds might be loose to explain what happens in experiments. For example, Figure 4b has significant decrease in empirical Lipschitz while maintaining clean accuracy, but there is no corresponding increase in adversarial robustness in Figure 3d. Indeed, the models with different widths at stage-3 all have similar PGD accuracies despite different empirical Lipschitz constant. There can be some other factors that is not captured here.

- The empirical evaluations on the effect of width, depth, and tradeoff parameter \lambda on adversarial robustness are very thorough. But at the end it still seems unsatisfying as there is no obvious trend or strong conclusions we can draw from it. If we change the dataset from CIFAR to another dataset the conclusions can become different. Or put it in another way, if we replace adversarial robustness (PGD accuracy) with some other quantities of interest (e.g., clean accuracy), we can still talk about the effect of width and depth and \lambda on clean accuracy. Without a clear trend or strong conclusion this type of empirical evaluations can only be treated as data points but not as insights into the problem.

- This paper is of good quality and very well-written, but the results presented do not seem to advance our understanding on the problem of adversarial robustness and neural architecture by a lot. This is a borderline paper to me.


**Time Spent Reviewing:**

3

---

> ### Author Response · Authors · 2021-08-10
> **Response to Reviewer DNQQ**
>
>
> Thanks for your thoughtful comments. We provide the following responses to your concerns.
>
>
> ---
> **Q1:** Figure 4b has a significant decrease in empirical Lipschitz while maintaining clean accuracy, but there is no corresponding increase in adversarial robustness in Figure 3d, other factors that are not captured.
>
> **A1:** We respectfully disagree that **there is no corresponding increase in adversarial robustness in Figure 3d**. Figure 3d demonstrates a clear trend where adversarial robustness is improving as the width is reduced: the PGD$^{20}$ robustness for width factors 10/8/6/4/2 are 52.04%/52.62%/53.57%/**54.28%**/53.95% respectively, i.e, there is a ~2% robustness increase. As we discussed at Lines 195-196 and 227-228, there is also a tradeoff: robustness drops when capacity becomes insufficient for proper learning. That is the reason why width reduction to $^{W}$10-10-2 slightly decreases the robustness of $^{W}$10-10-4, however, it is still notably higher than the vanilla configuration $^{W}$10-10-10. This is also reflected by the slight decrease of perturbation stability in Figure 4b. Compared to depth reduction, the width reduction is more noticable than depth reduction (see Figure 2c and Figure 3c), as we explained at Lines 218-222.
>
> In terms of other uncaptured factors, in our experiments, we have tried our best to make sure the width/depth is the only changing variable: all experiments were conducted with the same hardware infrastructure, GPU model, software version, random seed, and implementations. We have also repeated the width reduction experiments twice with different random seeds and double-checked the trend. The robustness difference according to PGD$^{20}$ is less than 0.2%. While we cannot be 100% sure to rule out all factors, we would like to point out that the consistent improvements against AA in Table 2 across 3 datasets (4 if including the ImageNet experiment in Appendix D.2), 4 types of DNNs, and 4 adversarial training methods indicate that our finding is indeed generic and can consistently improve robustness.
>
>
> ---
> **Q2:** If we change the dataset from CIFAR to another dataset the conclusions can become different. Or put it in another way, if we replace adversarial robustness (PGD accuracy) with some other quantities of interest (e.g., clean accuracy), we can still talk about the effect of width and depth and \lambda on clean accuracy. Without a clear trend or strong conclusion, this type of empirical evaluation can only be treated as data points but not as insights into the problem.
>
> **A2:** **Changing dataset.** Arguably, changing the searching dataset could lead to a different **optimal configuration**, for example, the different cell structures discovered in NAS on different datasets [1, 2, 3]. However, we would like to point out that the transferability of the discovery to improve other datasets or models is the key, for instance, in NAS, architectures discovered on CIFAR-10 are generally transferable to other datasets (such as ImageNet) [1, 2, 3]. The transferability and generality of our findings were verified by the consistent robustness improvement on different datasets with different models and training methods under both white-box and black-box (Appendix D.1) settings. Moreover, our ImageNet experiment in Appendix D.2 further shows that the same scaling rule can also be applied to improve the robustness of ResNet-50 trained using FastAT on ImageNet. This is also consistent with the results on CIFAR-10, CIFAR-10+500K, and CIFAR-100 in Table 2. We hope these results have provided necessary (if not sufficient) evidence to demonstrate the generality of our findings.
>
>
> **Replace PGD accuracy to clean accuracy**. We would like to point out that the focus of this work is **adversarial robustness**, rather than clean accuracy. And we believe PGD accuracy is a good indicator of adversarial robustness that has been widely adopted in the current literature. We cannot agree more with the reviewer that the discovery will be different if clean accuracy becomes the primary concern, however, that would fall into the scope of NAS. As we discussed at Lines 99-113, the discoveries in the two fields may not transfer to each other.
>
> Adversarial robustness is notably different from clean accuracy, in fact, they may be at odds with each other, as revealed in [4]. We believe our findings are new and notably different from the rule discovered with clean accuracy by EfficientNet [5]. Our theoretical analysis can also serve as a useful basis for robustness-oriented theoretical analysis.
>
>
> ---
> **Q3:** The results presented do not seem to advance our understanding on the problem of adversarial robustness and neural architecture by a lot. This is a borderline paper to me.
>
> **A3:** As we discussed in Appendix E.1, adversarial defense research is now at a stage where even 1%-2% improvement against AA is sufficient to propose a new defense method [6, 7]. Most defenses in the literature adopt the standard configurations or simply keep increasing network capacity (e.g. WRN-28-10 -> WRN-34-15 -> WRN-70-16). We show a similar level of improvement by a **simple reconfiguration** of the network, which we believe is beneficial to the community, especially when compared with using auxiliary data or sophisticated modifications to the training algorithm. Our discovered scaling rule can free researchers from computationally expensive architectural explorations in future studies or at a minimum, provide a useful rule of thumb for architectural decisions.
>
>
>
> ---
> [1] DARTS: Differentiable Architecture Search. ICLR 2019. \
> [2] Efficient Neural Architecture Search via Parameter Sharing. ICML 2018. \
> [3] ProxylessNAS: Direct Neural Architecture Search on Target Task and Hardware. ICLR 2019. \
> [4] Robustness May Be at Odds with Accuracy. ICLR 2019. \
> [5] EfficientNet: Rethinking Model Scaling for Convolutional Neural Networks. ICML 2019. \
> [6] Adversarial weight perturbation helps robust generalization. NeurIPS, 2020. \
> [7] Unlabeled data improves adversarial robustness. NeurIPS, 2019.

---

> > ### Author Response · Authors · 2021-08-25
> > **A follow up message**
> >
> > Dear Reviewer DNQQ,
> >
> > Thanks again for the valuable comments. Please let us know if anything is unclear. We truly appreciate this opportunity to improve our work and shall be most grateful for any feedback you could give to us.

---

> > > ### Comment · Reviewer_DNQQ · 2021-08-26
> > > **Thanks for the response**
> > >
> > > I would like to thank the authors for their clarifications. I agree with the authors' response that there is difference in adversarial robustness; my point was that the change in adversarial robustness is not proportionate to the change in empirical Lipschitz. It turns out I made a mistake with the y-scale in Figure 4b as the origin is not 0, there is only a 20% drop instead of 50% drop.
> > >
> > > The second part of my comments is about the significance and generalizability of this work. The authors have shown that increasing the width and depth do not necessarily increase adversarial robustness. But in practice to build a robust model for a new application, we still need to do some architecture search with width and depth, just like we do architecture search when training for clean accuracy. How could we make use of the insights provided in this paper to help with our architecture search?

---

> > > > ### Author Response · Authors · 2021-08-27
> > > > **Further clarifications**
> > > >
> > > > Thanks for your prompt reply! Please allow us to provide the following clarifications.
> > > >
> > > > **Not proportionate to the change in Figure 4.**
> > > > We use two y scales because the Empirical Lipschitz cannot be measured in percentage. Thus, it is expected that the changes are not proportional to the adversarial robustness. The Lipschitz measures the changes of the output with the change of the input for a given function. Lower Lipschitz indicates that the output (logits) changes are small between clean and adversarial samples. For adversarial robustness (accuracy on adversarial examples), it checks whether or not the maximum index of the logits (or probabilities) equals the correct label. The trend indicates that lower Lipschitz helps improve perturbation stability, which is a good indicator of robustness but not strictly proportional to robustness.
> > > >
> > > >
> > > > **How our insights can help with architecture search.**
> > > > For a new application, one may consider existing architectures like ResNet, WideResNet, VGG and DenseNet, for which our discovered robust configurations can be readily applied. We will also release the reconfigured versions of these architectures along with detailed instructions on how to apply our discovered rules to reconfigure a network.
> > > >
> > > > When NAS is considered, our findings can help design the overall network architecture. Reducing the width of the last stage can speed up the searching process while achieving better robustness. In our experiments, we have verified the robustness improvement on the final architecture discovered by DARTS. This indicates that incorporating our finding into the search space or the searching algorithm is also promising. Future NAS work may also consider using a separate search space, search strategy, or cell structure for the last stage of the network.
> > > >
> > > > Thanks again for your feedback. Please kindly let us know if further clarification is needed.

---

### Decision · Program_Chairs · 2021-09-27

**Decision:**

Accept (Poster)

**Comment:**

This paper studies the impact of network width and depth on the robustness of a model through adversarial training. Different from previous works that looked into the impact by varying the overall width and depth of a network, the authors perform stage-level experiments in which they modify the width and depth of different stages within a model. The findings, which indicate that decreasing the depth and width at the last stage can lead to higher robustness while decreasing depth and width at earlier stages of the network hurts robustness, are useful to the community. However, the upper bound on the Lipschitz constant proposed in this work does not seem to explain the empirical results well. We encourage the authors to discuss the consistency between the theoretical and empirical results more if the paper is finally accepted.